# The Show Must Go On: A Snapshot of Italian Academic Working Life during Mandatory Work from Home through the Results of a National Survey

Chiara Ghislieri [1], Domenico Sanseverino [1,*], Tindara Addabbo [2], Vincenzo Bochicchio [3], Rosy Musumeci [4], Ilenia Picardi [5], Patrizia Tomio [6], Gloria Guidetti [7] and Daniela Converso [1]

1   Department of Psychology, University of Turin, Via Verdi, 10, 10124 Turin, Italy; chiara.ghislieri@unito.it (C.G.); daniela.converso@unito.it (D.C.)
2   Marco Biagi Department of Economics, University of Modena and Reggio Emilia, Via Jacopo Berengario, 51, 41121 Modena, Italy; tindara.addabbo@unimore.it
3   Department of Humanities, University of Calabria, Via P. Bucci 18/C, 87036 Rende, Italy; vincenzo.bochicchio@unical.it
4   Department of Cultures, Politics and Society, University of Turin, Lungo Dora Siena, 100A, 10153 Turin, Italy; rosy.musumeci@unito.it
5   Department of Political Sciences, University of Naples Federico II, Via Rodinò, 22/a, 80128 Naples, Italy; ilenia.picardi@unina.it
6   Diversity and Disability Manager, University of Trento, Via Calepina, 14, 38122 Trento, Italy; patrizia.tomio@unitn.it
7   Department of Psychological, Health and Territorial Sciences, University G. D'Annunzio of Chieti-Pescara, Via dei Vestini, 31, 66100 Chieti, Italy; gloria.guidetti@unich.it
*   Correspondence: domenico.sanseverino@unito.it

**Abstract:** During the COVID-19 pandemic, universities worldwide have provided continuity to research and teaching through mandatory work from home. Taking into account the specificities of the Italian academic environment and using the Job Demand-Resource-Recovery model, the present study provides, through an online survey, for the first time a description of the experiences of a large sample of academics (N = 2365) and technical and administrative staff (N = 4086) working in Italian universities. The study analyzes the main differences between genders, roles or work areas, in terms of some job demands, recovery experiences, and outcomes, all important dimensions to achieve goals 3, 4, and 5 of the 2030 Agenda for Sustainable Development. The results support the reflections on gender equality measures in universities and provide a general framework useful for further in-depth analysis and development of measures in order to improve well-being (SDG 3), quality of education (SDG 4), and gender equality (SDG 5).

**Keywords:** mandatory work from home; Job Demand-Resource-Recovery model; Italian academia; gender differences; Sustainable Development Goals (SDGs) and universities; academic sustainability

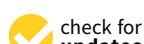



## 1. Introduction

Mandatory work from home (Kniffin et al. 2021; Ghislieri et al. 2021b) at universities worldwide allowed academic organizational life to continue during the COVID-19 pandemic and ensured continuity of research and teaching, despite some well-known limitations (such as lack of access to laboratories and lack of informal interpersonal relationships). Remote academic work has prevented students from having to interrupt their careers, thus also helping to compensate for the lack of social and activities time, despite heavy study loads and not always satisfactory support from faculty (Aristovnik et al. 2020). However, how was this experience lived by the academics and the technical and administrative staff (TAS)[1], in terms of some important variables modulating the organizational experience?

Taking into account the specificities of the Italian academic context (Ghislieri et al. 2014; Converso et al. 2019) and using as main reference the Job Demand-Resource-Recovery model (Kinnunen et al. 2011), the present study provides, for the first time, a snapshot of the experiences of a large sample of academics and TAS working in Italian universities.

In particular, the paper analyzes the main differences in gender (for academics and TAS), role (academics) and area of work (TAS), with respect to some demands (workload, cognitive demands, off-hours technology assisted job demands, workaholism, work–family conflict, remote working evaluation), recovery experiences (detachment, relaxation, mastery, and control) and emotional exhaustion as outcomes.

This paper is also embedded in the field of psychosocial aspects of sustainability: the aim of this approach is to promote healthier organizations and to balance the necessary changes in working conditions with the maintenance of adequate human well-being (Di Fabio 2017; Di Fabio and Rosen 2018; Molino et al. 2019). Citing the recommendation to include multiple disciplines in achieving sustainability goals (Findler et al. 2019), other contributions have highlighted the importance of interpersonal, social, and well-being dimensions as key elements in defining sustainable academic organizations (Gamage et al. 2022). Therefore, this study's approach is consistent with Goal 3 of the 2030 Agenda: by addressing the organizational life of universities, the study also lends itself as an important node for promoting quality education (Goal 4), as the well-being of academic staff is reflected in the quality of higher education. Finally, the focus on gender differences is consistent with the gender equality dimension (Goal 5).

This picture of some aspects of Italian academic working life at a specific point in time, namely the second lockdown since the beginning of the pandemic (winter 2020), unfortunately cannot be compared with other descriptions of the same indicators due to the lack of previous similar studies. This is one of the main limitations of this work, which, however, offers a broad look at the system of demands and resources of academics and TAS, with particular attention to the evaluation of mandatory work from home and differences based on gender and role. The study fills a gap in the availability of descriptive information on specific indicators of the quality of academic life and thus provides a useful framework for in-depth analysis of the relationships between the observed variables.

### 1.1. The Italian "University Job"

The Italian "university job" today is the result of a decade of implementation of the most recent university reform, which has brought significant changes in the Italian academic institution. Law 240 (30/12/2010), known as the Gelmini reform, introduced important transformations in academia, both for academics and for TAS.

Previous studies, shortly after the COVID-19 pandemic, have shown how it led to an increase in the demands on management and administration, with an increase in the number of steps, including bureaucratic ones, to certify continuous evaluation processes, both on the teaching and research side; this change has had an impact on the work of both academics and TAS (Ghislieri et al. 2014). In particular, the TAS were asked to contribute more and more to develop new knowledge and skills (e.g., in fundraising and evaluation processes or in transversal and digital skills), often without adequate training tools (i.e., at "zero cost", the mantra of the Gelmini reform). Even the academic profession, once considered prestigious, remunerative, socially recognized, and highly desirable (Rostan 2011; Ghislieri et al. 2014), has changed in the last two decades. The positive image of academic work has progressively faded, not only in Italy (Winefield et al. 2003), and this phenomenon has been associated with other aspects such as reduced funding, a decisive and protracted reduction in access to academic careers, an increase in precariousness, the blocking of turnover and salary jumps, limited promotion opportunities, and a significant increase in the bureaucratic burden associated with the management of teaching, mostly under the condition of limited resources.

Although it is not possible to make a systematic comparison between the data collected in this survey and previous data, the analysis of studies in the academic field prior to

the pandemic period provides us with a framework that helps to highlight some pre-existing critical issues. For example, a study by Ghislieri et al. (2014) at a medium-sized university observed the differences between academics and TAS in the relationship between demands-resources and job satisfaction, highlighting the fundamental role of autonomy for academics and supervisory support for TAS. Moreover, work–family conflict was found to be negatively correlated with satisfaction, especially for academics. More recently, a study by Converso et al. (2019) found significant levels of workaholism among Italian academics: the study highlights that work engagement and workaholism can be considered as the positive and negative sides, respectively, of the high work investment of academics in Italy, pointing to the importance of promoting work resources (meaning of work, rewards) and controlling demands (especially work overload).

Using a person-centered approach, another study (Guidetti et al. 2020) identified different occupational profiles related to well-being in a sample of Italian university professors, defined by different levels of work engagement, emotional exhaustion, workaholism, and job satisfaction: detached (30.4%), exhausted-workaholic (21.4%), engaged-workaholic (26.3%), engaged-satisfied (21.8%). Whether engaged or detached, workaholism appears to be an important concern in academic work dynamics. The analyses also revealed significant differences between the different profiles in terms of perceptions of some specific job demands in the academic context, distinguishing between demands perceived as hindering (hindrance demands) and demands perceived as challenging (challenge demands).

### 1.2. Roles and Gender Distribution in Academia

As mentioned above, academic careers in Italy are regulated by Law 240/2010, which defines four different roles: Full Professor, Associate Professor, and two types of fixed-terms research contracts, an A-type (three years, possibly renewable for 2 years, without tenure track; RTD-A) and a B-type (with tenure track, with national scientific qualification; RTD-B); the two new types of fixed-term contracts replaced the former open-ended contract of the Assistant Professor, the so-called "Ricercatore Universitario" (RU), which can be considered a Senior Assistant Professor. The RU from before the Gelmini reform deserve a separate mention; although there are no new hires for this role, there are still a large number of RU in universities. Although institutional engagement and governance demands are higher in senior positions, all academic staff are required to engage in research, teaching, public engagement, and faculty bureaucracy.

According to 2020 data from the Ministry of University and Research (Ministero dell'Istruzione 2020), shown in Figure 1, the gender gap in academia still exists and widens in senior positions, with the largest gap among full professors. While there is almost no gender gap for RU, the peculiar nature of this role and the difficulties in career progressions, due to the redefinition of role advancement, do not necessarily indicate better career prospects for women. The situation is reversed for TAS, of which women constitute 60.1%; however, as noted in two recently published MUR documents (Ministero dell'Istruzione 2021a, 2021b), there are both vertical and horizontal gender asymmetries. In 2019, only 38% of technical staff were women, compared to 74% of administrative staff; in 2020, only 41.2% of administrative directors were women.

Concerning academics, the underrepresentation of women in top positions is linked to the underrepresentation of women in decision-making positions, highlighting the issue of the glass ceiling, as the She Figures 2021 (European Commission 2021b) report shows. Despite the progress made in gender equality, equality tools, and policies implemented, the phenomenon of cultural sexism persist in Italian academia (Savigny 2014) and women are still promoted less frequently than their male counterparts, and, as some studies show, are less likely to access the most "secure" academic positions. Picardi's (2019) analysis shows that women are less likely to occupy the first secure position in the academic career, taking into account data from the Italian Ministry of Education, University, and Research, confirming a reinforcement of gendered selection in access to the academic profession after the implementation of the Gelmini reform.

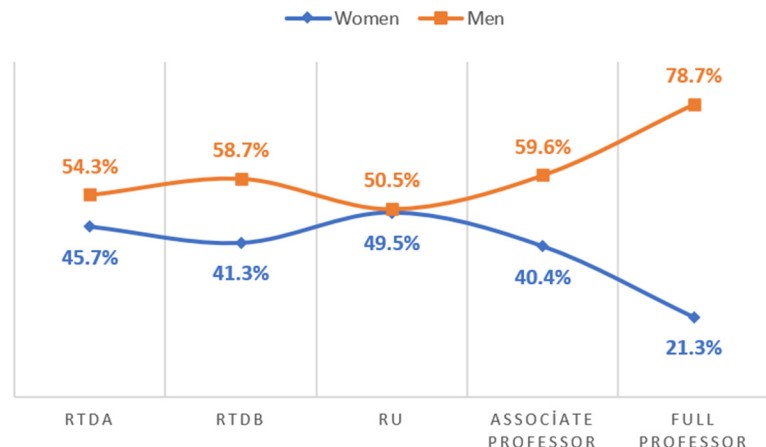

**Figure 1.** Gender distribution of academic roles in Italy in 2020.

Other studies confirm that the path to gender equality is slow and nonlinear (Gaiaschi and Musumeci 2020), noting that the (slow) growth of the feminization of the academic composition is the result of a demographic process and not of performance policies that contribute to narrowing the gap. Looking at the recruitment process, we find that with the Gelmini reform, women's access to the RTD-B role has been limited and that the possibility of obtaining a full professorship has remained essentially the same. Other studies, mainly related to bibliometric disciplines, had already highlighted that the lower likelihood of women advancing to higher positions in academic careers, despite systematically better scholastic and academic performance (Murgia and Poggio 2011), is neither due to lower scientific productivity nor to negative self-selection (Filandri and Pasqua 2019).

To understand the phenomenon of gender segregation in academia, it is well known that it is necessary to consider a number of possible causes (Murgia and Poggio 2018). Studies by Heijstra et al. (2016, 2017), also using Italian data, have shown a different impact of domestic work on the working conditions of women and men in academia at the beginning of their careers, as well as a different impact of "academic domestic work", understood figuratively as all that academic service work within the institution that is characterized by little recognition and enhancement for career purposes, which requires a large investment of time and energy. Given that the measures taken to contain the pandemic disproportionately impacted women, for whom the burden of care became even heavier (European Commission 2021a; Manzo and Minello 2020), it is of particular interest to examine the gender differences in academia during the pandemic.

### 1.3. Work from Home

Allen et al. (2015) predicted that remote working would be a useful tool for organizational continuity in the event of an epidemic. Notwithstanding this awareness, most Italian (and other) organizations were clearly not prepared for this eventuality. Despite pre-pandemic government calls to experiment with "smart" forms of remote working in public administration (smart working or agile working, regulated by Capo II of Legge 22 maggio 2017, n.81; the first term is used exclusively in Italy), few public administrations and especially, few universities, had moved in the indicated direction. Referring, for example, to TAS, in a national survey (Ghislieri et al. 2021a), only 8 of the 44 participating universities provided information on remote working before the pandemic, reporting a figure of around 16% of women and 10% of men involved in experimental remote working. In the same survey, 21 universities reported data on staff working remotely during the initial lockdown, with 67% of women and 62% of men working remotely. There were also shortcomings in remote teaching: academics were generally inadequately qualified and unfamiliar with technological tools; in addition, universities generally offered little or no

technical support for remote teaching, including sparse development of web conferencing tools or online platforms for sharing materials and other resources.

In summary, during the first lockdown the Italian academy was in no way prepared to work remotely; knowledge and skills were developed during practice in the adversity of the situation. Between the first and the second lockdown, most organizational efforts involved building the conditions for a return to presence; important and necessary efforts, but again, only in rare cases did they involve training investments aimed at developing skills and competences for remote work, both in consideration of a physiological evolution of this work approach and to be prepared for an eventual new lockdown (Ghislieri et al. 2021a). The Three-Year Plan for Information Technology in Public Administration 2020–2022 (Agenzia per l'Italia Digitale 2020) has revealed a strong delay in the digitization process of the Italian public administration. The lockdown experience has elicited harsh shortcomings in technological tools, besides the need to redesign processes and promote a user-centered administration through the assignment of objectives and the monitoring of results. This delay helps to illustrate the difficulties that have been highlighted in studies on remote working during the emergency. However, these complications may provide a starting point for potential future interventions.

In the last two years, the topic of remote working has been widely discussed both in academia and in public discourse; in Italy, many organizations made a massive switch to remote work starting in March 2020 in order to contain the spread of COVID-19 and ensure continuity of work. However, this transition was quite sudden and resembled a large, unprogrammed experiment rather than a carefully planned work transition. This was particularly true in Italy, where remote working was not widespread and actually encompassed two different types of work arrangements (Ghislieri et al. 2021b).

Before the emergency, 570,000 people worked from home, mainly in large companies, while only 16% of public administrations had set up remote work projects, although there was much normative support in the past, including direct promotion by the government. Following the Decree of the President of the Council of Ministers promulgated on 11 March 2020 and the directives of the Ministry of Public Administration, which effectively extended remote working to all personnel bypassing the necessary trial period, the rate of remote workers in public administrations reached 68%. The legal situation is currently evolving and new measures are still being considered.

While the Italian government and public discourse in general have referred to the emergency remote work as "lavoro agile" or "smart working", the term "mandatory work from home" (MWFH), proposed by Kniffin et al. (2021), seems more appropriate. Specifically, we refer to a hybrid arrangement: especially in the early months, most people worked exclusively from home throughout the week, but often with their own devices and highly variable job verification and evaluation procedures that were either more traditional, such as time spent connected, or objectives-driven, which was more akin to agile working. Moreover, the pervasiveness of domestic care work, combined with social and professional isolation, could impair recovery experiences or at least reduce available options, which in turn negatively affects quality of life and work.

Regardless of the Italian context, there are numerous critical issues to consider regarding remote working, both in terms of the ongoing emergency and in terms of the so-called "new normal" (Kniffin et al. 2021; Rudolph et al. 2021). Several studies have examined the positive and negative aspects of remote work both before the COVID-19 emergency (Allen et al. 2015) and during (Barbuto et al. 2020; Mustajab et al. 2020), focusing on changes in work demands and the work–life interface, technostress, work quality and performance, and well-being (Fana et al. 2020; Hamouche 2020; Galanti et al. 2021; Molino et al. 2020; Oksanen et al. 2021; Toscano and Zappalà 2020; Aczel et al. 2021).

Specific to academia, Kaiser et al. (2021) observed a strong relationship between job demands and burnout in a sample of 236 employees at a Norwegian university, with autonomy being the most important resource for predicting commitment and reducing burnout. In addition, a study conducted with 300 employees of two international private

universities in Thailand (Charoensukmongkol and Phungsoonthorn 2020a) while working remotely found that perceived uncertainties mediated the negative relationship between crisis communication and emotional exhaustion; informal communication may play a role in formal communication dynamics to reduce uncertainty during a crisis. Charoensukmongkol and Phungsoonthorn (2020b) also point out that the negative effect of supervisor support on employees' perceived uncertainties is only present in universities characterized by low intransigence, highlighting the importance of the overall work climate in universities. Deryugina et al. (2021), in their analysis of researchers' time allocation (identified by having written at least one academic paper in the last 5 years), show a decrease in working hours, especially in research, and an increase in unpaid care work, especially for mothers with younger children.

*1.4. Job Demands-Resource-Recovery Model*

The Job Demands-Resources model (JD-R; Bakker and Demerouti 2017) is a theoretical proposal that has been widely validated empirically and aims to provide a general model, adaptable to specific contexts, to explain both the motivational process (and thus work engagement) and the process of health degradation (burnout).

Recognizing the importance of recovery experiences, Kinnunen et al. (2011) proposed the Job Demands-Resource-Recovery model (JD-R-R; Kinnunen et al. 2011), a further elaboration of the JD-R model, that assumes that recovery experiences act as mediators in the relationships between job demands, resources, and the two JD-R processes. Indeed, recovery experiences replenish and create resources that are fundamental to well-being, by promoting pleasant activation (vigor) and helping to endure unpleasant deactivation (fatigue) (Bennett et al. 2018).

In this study, we focused on both academics and TAS in universities. According to the JD-R-R model, we examined recovery experiences along with various job demands, namely workload, cognitive demands, and off-hours technology-assisted job demands (off-TAJD); we also included workaholism and work–family conflict, which can be considered both outcomes and demands. In addition, we considered perceptions of advantages and disadvantages of remote working. Finally, we examined emotional exhaustion, an outcome related to the process of health degradation.

Job demands have been defined as "those physical, psychological, social, or organizational aspects of the job that require sustained physical and/or psychological (cognitive and emotional) effort or skill and are therefore associated with certain physiological and/or psychological costs" (Bakker and Demerouti 2007, p. 312). Given the nature of work in universities, which involves information seeking and processing, problem solving, and decision making, we chose to analyze workload and cognitive demands, demands that can lead to energy depletion, emotional exhaustion, and other health problems (Bakker and Demerouti 2017), such as burnout, which in turn prevents optimal use of cognitive resources and thus limits cognitive functioning and consequently job performance (Lemonaki et al. 2021).

The recovery process is fundamental to restoring energy depleted by job demands to pre-stressors levels (Meijman and Mulder 1998). Sonnentag and Fritz (2007) describe four different recovery experiences, i.e., the activities outside of work through which recovery processes occur. These activities can be traced to the effort-recovery (Meijman and Mulder 1998) and conservation of resources (Hobfoll and Wells 1998) models: (1) psychological detachment from work; (2) relaxation, a calm state of low arousal; (3) mastery, which refers to activities other than work that provide challenging experiences; and (4) control over one's leisure activities and time.

The literature has highlighted the positive effect of recovery on well-being (e.g., Kinnunen et al. 2011) and performance (e.g., Sonnentag and Bayer 2005), while also highlighting its role as a mediator in the relationship between work–family conflict, psychological tension, and life satisfaction (Moreno-Jiménez et al. 2009), between workload and work–family conflict (Molino et al. 2016), hindrance, and challenges demands with vigor

and fatigue (Bennett et al. 2018) and, specifically during remote work, between off-TAJD and emotional exhaustion (Dolce et al. 2020).

However, the peculiar nature of MWFH could present additional barriers to proper recovery, such as the overuse of technology, which was often the only way to complete work-related tasks; also, given the prevalence of an always-on approach (Derks et al. 2015; Ghislieri et al. 2018; McDowall and Kinman 2017), this context could further increase technostress (Tarafdar et al. 2007). Specifically, we considered two technostress creators, technological invasion and overload, contained in off-TAJD, i.e., the perceived request to use technological devices to respond to work demands outside of regular work hours. Constant availability and further blurring of life and work domains can also worsen the work–life balance through work intensification (Bordi et al. 2018; Derks and Bakker 2014; Molino et al. 2020; Yun et al. 2012), affect job productivity and job and life satisfaction (La Torre et al. 2019), and impair recovery (Dolce et al. 2020).

Increased job demands, particularly workload, work intensification, and technology use, have also been linked to workaholism (Andreassen et al. 2019; Molino et al. 2020; Spagnoli et al. 2019), which is directly or implicitly promoted in some organizations (Molino et al. 2019). Clark et al. (2020) noted that workaholism is a multidimensional construct articulated in four dimensions: an internal compulsion to work, persistent and uncontrollable thoughts about work, feeling negative emotions when not working or when prevented from working, and excessive working beyond expectations and requirements. While the four dimensions refer to different aspects (motivational, cognitive, emotional, and behavioral), Tóth-Király et al. (2021) argue that the cognitive and motivational aspects of workaholism distinguish between excessive working, i.e., high work investment, and compulsive working, and that the risks associated with workaholism mainly stem from the second aspect. Workaholism is a state of high activation, similar to engagement, but associated with unpleasant emotions (Bakker and Oerlemans 2011). It could be argued that a tendency to work hard, coupled with disruptive work contexts, e.g., destructive leadership (Molino et al. 2016, 2019), personal traits such as perfectionism (Falco et al. 2013), and an increase in technology that allows one to work incessantly, anytime, and anywhere, could lead workers to develop the compulsion associated with workaholism and thus engage in workaholic behaviors.

Spagnoli and Molinaro (2020) elaborated on this last theme, stating that during the lockdown, workers suffered from the sudden deprivation of their usual working conditions and may have experienced an increase in negative workaholic emotions, which, as mentioned earlier, is related to emotional exhaustion. Moreover, workaholism impairs the natural recovery process (Bakker et al. 2013; Molino et al. 2018), also because workaholics who experience negative emotions at the end of the workday tend to engage in further work activities (Van Wijhe et al. 2013). Several papers have documented the consequences of workaholism, other than emotional exhaustion (Gillet et al. 2017): increased stress levels, less job and life satisfaction, and increased work–family conflict (Clark et al. 2016).

The work–family interface, particularly in the context of remote working, is a central variable for studying work-related well-being and the effectiveness of flexible work arrangements. Work–family conflict (WFC) stems from the role strain hypothesis (Goode 1960) in the context of role theory (Merton 1957), which posits that managing multiple roles, each of which requires an investment of energy and time, can lead to inter-role conflict that imply a discordance or incompatibility between work and life domains and their associated demands (Greenhaus and Beutell 1985). The context of the pandemic and the resulting temporary interruptions of educational and school services had obvious repercussions on the work–family interface, including social isolation, misuse of technology, difficulty in separating domains, and increased work demands (Chung et al. 2020). Both before and during the pandemic, it was highlighted that work-related demands contributed most to WFC, which in turn was related to other outcomes, such as emotional exhaustion, poorer job performance, increased turnover intentions, and lower job and life satisfaction (Allen et al. 2020; Amstad et al. 2011; Ghislieri et al. 2012; Vaziri et al. 2020).

Considering these facts, most studies have emphasized the negative sides or disadvantages of home-based work. Nevertheless, even during the pandemic period, past literature (Ipsen et al. 2021) focused on how people perceive working from home, both in terms of advantages and disadvantages. This could be one aspect to guide and manage the organization of remote workers and to take effective action. Based on previous research, in addition to the negative aspects such as role ambiguity, social isolation, deterioration of social support from supervisors, and challenges to work–life balance, the positive aspects of working from home should be considered. Previous studies have noted potential benefits such as increased productivity, better work–life balance, less stress from telecommuting, or more control over one's work and daily health routines (Anderson et al. 2015).

In this sense, and according to the JD-R model, the disadvantages of WFH can be considered as job demands that negatively affect employee's well-being, while the perceived advantages can be defined as job resources that protect against negative work- and health-related outcomes associated with WFH. According to Ipsen et al. (2021), there is a need to gain a deeper understanding of how people experience WFH and "how they perceive the advantages and disadvantages of their new situation" (p. 3). Accordingly, this study, along with the previously described dimensions, will provide insights into the perceived advantages and disadvantages of remote work.

## 2. Materials and Methods

### 2.1. Procedure

The study is part of a broader research-intervention project of the National Conference for Equality in Italian Universities. The Conference gathers representatives of the university committees working on issues of equal opportunities and well-being to promote collaboration among faculty in the areas of gender equality, work–life balance, well-being, and inclusion.

Following data collection, national findings were presented to member universities through a presentation and discussion workshop that identified current needs and potential general interventions. Consistent with an intervention-based research approach, each member later received a more detailed report explaining each examined area.

### 2.2. Participants

Participants completed an online self-report questionnaire on the Limesurvey platform; data were collected between December 2020 and March 2021. Response rates varied widely across universities. In the TAS sample, the average response rate was 26.57% (SD = 12.97%; Min = 4.33%, Max = 45.37%), while in the academic sample it was 22.08% (SD = 13.07%; Min = 4.79%, Max = 48.51%).

Prior to analysis, we performed preliminary data cleaning. For the TAS survey, we excluded 414 cases with more than 10% missing responses to the study variables (N = 3672); the same was done for the academics survey, where we excluded 190 cases (N = 2175).

The majority of the TAS sample was female (N = 2600; 72%); the mean age was 48.02 years (SD = 9.10) and ranged from 21 to 67 years. In the sample, 38.7% worked at large universities (20,000 to 40,000 enrolled students), 26.8% at medium universities (10,000 to 20,000 enrolled students), 16.1% at polytechnics, 14.5% at "mega" universities (more than 40,000 enrolled students), and 3.9% at small universities (up to 10,000 students). The majority of participants had a permanent contract (91.5%) and worked full time (88.1%), with a mean seniority of 17.79 years (SD = 10.44). On average, participants worked 36.94 h per week (SD = 8.26). To better protect individual privacy, we did not ask for information that could potentially be used to identify participants but examined the work area rather than the individual role to favor future interventions related to remote work.

Most individuals were employed in educational services (30.7%), followed by administration (18.6%) and research (18.3%). The other third of the sample was employed in ICT (9.5%), human resources (8.2%), finance (5.3%), internationalization (4.4%), logistics and maintenance (3.8%), and legal affairs (1.1%).

The academic sample was balanced in terms of gender (men N = 1086, 50.5%; women N = 1063, 48.9%). The mean age was 50.74 years (SD = 9.31) and ranged from 26 to 70 years. Moreover, 45.4% worked at large universities, 27.1% at medium universities, 17.1% at polytechnics, 8.1% at "mega" universities, and 2.3% at small universities. The majority consisted of Associate Professors (AP; 42.3%), followed by Full Professors (FP; 23.5%), permanent researchers (RU; 14.4%), and temporary researchers (RTD-B: 11%; RTD-A: 8.8%). The research did not involve procedures that might affect participants' psychological or social well-being, in accordance with the Declaration of Helsinki (World Medical Association 2013); all participants gave informed consent and were assured of data anonymity. The cover letter indicated the aim of the study, instructions for completing the questionnaire, the voluntary and unpaid participation, and information on data processing.

*2.3. Data Analysis*

We performed analyses of variance (ANOVA) for both samples to assess differences in study variables between women and men. For the TAS sample, we also tested for significant differences across nine work areas, while for the academic sample, we compared means across five academic levels. When comparing more than two groups, we conducted post-hoc tests (Tukey post-hoc test for variables that met the assumption of homogeneity of variance and Game–Howell for the others). All analyses were performed using IBM SPSS, version 26.

*2.4. Measures*

Workload was measured with 2 items (Bakker et al. 2004) on a Likert scale from 1 ("never") to 5 ("always"). An example item is "I have to work under pressure". The Guttman split-half coefficient was 0.82 for both samples.

Cognitive demands were assessed using 2 items (Bakker et al. 2003) already used in other Italian studies (e.g., Molino et al. 2019), with a Likert scale ranging from 1 ("never") to 5 ("always"). An example item is "My work requires constant attention". The Guttman split-half coefficient was 0.72 for the TAS sample and 0.58 for the academic sample.

Off-TAJD was measured with four items adapted from the Ghislieri et al. (2017) scale. Participants were asked to indicate how often they felt their organization required them to work beyond the agreed-upon hours using technology. Participants were asked to rate the statements on a five-point Likert scale ranging from 1 ("never") to 5 ("always"). An example item is "How often does your organization require you to answer phone calls and emails on weekends and/or non-workdays?". Cronbach's alpha was 0.92 in the TAS sample and 0.82 in the academic sample.

Workaholism was assessed with the Italian adaptation of the Bergen Work Addiction Scale (BWAS; Andreassen et al. 2012; Italian version Molino 2013). The measure consists of 7 items on a 5-point Likert scale ranging from 1 ("Never") to 5 ("Always"). An example item is "Thinking about the last week, how often did you give less priority to hobbies, leisure activities, and physical activity because of your work?". Cronbach's alpha was 0.79 in the TAS sample and 0.74 in the academic sample.

Work–family conflict was measured using the Italian adaptation (Colombo and Ghislieri 2008) of Netemeyer et al.'s (1996) scale, which consists of 5 items on a Likert scale ranging from 1 ("never") to 5 ("always"). An example item is "The amount of time my job requires makes it difficult for me to meet my family obligations." An item measuring conflict in the family to work direction was added to this scale. Cronbach's alpha was 0.86 in the TAS sample and 0.84 in the academic sample.

Recovery was measured using 12 items (Sonnentag and Fritz 2007) previously used in other Italian studies (Ghislieri et al. 2021b). Participants were asked to rate statements on a five-point Likert scale ranging from 1 ("strongly disagree") to 5 ("strongly agree"). Four dimensions define the factor structure of this scale: psychological detachment, relaxation, mastery, and control; each dimension was measured with three items, except for mastery, which included four items. Participants were asked to reflect on their after-work activities;

some sample items are "I forget about work" (psychological detachment), "I do relaxing activities" (relaxation), "I seek intellectual challenges" (mastery), and "I determine my own schedule" (control). Cronbach's alpha coefficients for TAS were 0.92 (detachment and relaxation), 0.84 (mastery), and 0.82 (control); for academics, they were 0.89 (detachment), 0.91 (relaxation), 0.77 (mastery), and 0.75 (control).

Remote work disadvantages were assessed using 12 ad hoc items on a 5-point Likert scale. Participants were asked to rate on a scale of 1 ("Not at all") to 5 ("Totally") how much some negative aspects of remote working affected their well-being and/or daily schedule. Cronbach's alpha was 0.89 in the staff sample and 0.88 in the academic sample.

Remote work advantages were assessed using 9 ad hoc items on a 5-point Likert scale. Participants were asked to rate on a scale of 1 ("Not at all") to 5 ("Totally") how much some positive aspects of remote working affected their well-being and/or daily schedule. Cronbach's alpha was 0.89 in both samples. Questions about the advantages and disadvantages were based on existing knowledge about remote work from before the pandemic and from the current situation (e.g., Kurland and Bailey 1999; Fonner and Roloff 2010; Cooper and Kurland 2002; Organisation for Economic Co-operation and Development 2021) and aimed to analyze aspects related to the home–work interface, work-related productivity, organization, quality of communication, and relationships with physically distant colleagues.

Emotional exhaustion was assessed using eight items from the Oldenburg Burnout Inventory (OLBI) (Demerouti et al. 2010) on a five-point Likert scale ranging from 1 ("Strongly disagree") to 5 ("Strongly agree"). Participants were asked to rate statements such as "during my work, I often feel emotionally drained". Cronbach's alpha was 0.82 in both samples.

## 3. Results

### 3.1. TAS Sample

Table 1 shows the means and standard deviations of the variables for the male and female TAS subsamples, followed by the ANOVA results. The only variables that did not show a significant difference between genders were detachment, off-TAJD, and remote work advantages. Women reported higher levels of work demands, workaholism, emotional exhaustion, and WFC, while reporting lower levels of recovery; finally, on average, women reported a slightly higher impact of remote work disadvantages.

**Table 1.** Means, standard deviations, and ANOVA for the TAS sample by gender.

| | Women | | Men | | | |
|---|---|---|---|---|---|---|
| | **M** | **SD** | **M** | **SD** | **F** | **p** |
| Workload | 3.58 | 0.98 | 3.30 | 0.96 | 57.72 | 0.000 |
| Cognitive demands | 3.94 | 0.79 | 3.81 | 0.79 | 20.67 | 0.000 |
| Off-TAJD | 2.52 | 1.18 | 2.46 | 1.15 | 1.76 | 0.185 |
| Workaholism | 2.30 | 0.85 | 2.18 | 0.82 | 13.58 | 0.000 |
| WFC | 2.48 | 0.90 | 2.30 | 0.88 | 29.12 | 0.000 |
| Detachment | 2.81 | 1.16 | 2.85 | 1.15 | 1.33 | 0.250 |
| Relaxation | 3.22 | 1.13 | 3.60 | 0.99 | 91.76 | 0.000 |
| Mastery | 3.10 | 1.02 | 3.43 | 0.93 | 75.59 | 0.000 |
| Control | 3.16 | 1.05 | 3.27 | 1.05 | 8.30 | 0.004 |
| Remote work disadvantages | 2.68 | 0.95 | 2.53 | 0.88 | 18.92 | 0.000 |
| Remote work advantages | 3.61 | 0.97 | 3.63 | 0.92 | 0.09 | 0.764 |
| Emotional exhaustion | 2.82 | 0.76 | 2.64 | 0.75 | 44.11 | 0.000 |

Means, standard deviations, and ANOVA results for the different work areas are shown in Table 2, while the post-hoc tests are shown in Table 3. No significant differences were found for control, remote work advantages, and WFC in the ANOVA. Moreover, a post hoc test showed that there were no significant differences for workload and detachment either.

Table 2. Means, standard deviations, and ANOVA for the TAS sample by work areas.

| | Human Resources | | ICT | | Logistics and Maintenance | | Research | | Educational Services | | Internaz. | | Finance | | Administration | | Legal Affairs | | | |
|---|---|---|---|---|---|---|---|---|---|---|---|---|---|---|---|---|---|---|---|---|---|
| | M | SD | M | SD | M | SD | M | SD | M | SD | M | SD | M | SD | M | SD | M | SD | *F* | *p* |
| Workload | 3.55 | 0.96 | 3.39 | 0.96 | 3.47 | 1.06 | 3.46 | 0.97 | 3.52 | 0.99 | 3.68 | 0.90 | 3.70 | 0.98 | 3.44 | 0.98 | 3.75 | 0.97 | 2.58 | 0.008 |
| Cognitive demands | 3.91 | 0.76 | 3.99 | 0.78 | 3.77 | 0.86 | 3.98 | 0.78 | 3.84 | 0.81 | 3.88 | 0.78 | 4.17 | 0.72 | 3.87 | 0.80 | 4.28 | 0.71 | 5.45 | 0.000 |
| Off-TAJD | 2.24 | 1.13 | 2.51 | 1.10 | 2.95 | 1.10 | 2.74 | 1.13 | 2.44 | 1.16 | 2.15 | 1.10 | 2.43 | 1.20 | 2.37 | 1.16 | 2.60 | 1.16 | 9.45 | 0.000 |
| Workaholism | 2.16 | 0.87 | 2.07 | 0.75 | 2.36 | 0.75 | 2.31 | 0.81 | 2.30 | 0.87 | 2.29 | 0.85 | 2.37 | 0.89 | 2.28 | 0.86 | 2.28 | 0.65 | 3.13 | 0.002 |
| WFC | 2.36 | 0.94 | 2.31 | 0.85 | 2.52 | 0.81 | 2.46 | 0.83 | 2.44 | 0.91 | 2.33 | 0.94 | 2.54 | 0.99 | 2.39 | 0.92 | 2.51 | 0.87 | 1.70 | 0.000 |
| Detachment | 2.92 | 1.19 | 2.92 | 1.18 | 2.94 | 1.11 | 2.67 | 1.10 | 2.78 | 1.14 | 2.93 | 0.11 | 2.88 | 1.22 | 2.88 | 1.20 | 2.97 | 1.04 | 2.52 | 0.010 |
| Relaxation | 3.44 | 1.10 | 3.55 | 1.07 | 3.37 | 1.10 | 3.23 | 1.06 | 3.26 | 1.12 | 3.34 | 1.01 | 3.35 | 1.22 | 3.33 | 1.16 | 3.15 | 0.92 | 2.79 | 0.004 |
| Mastery | 3.28 | 0.95 | 3.34 | 0.99 | 3.25 | 0.94 | 3.15 | 1.01 | 3.09 | 1.00 | 3.37 | 0.94 | 3.15 | 1.14 | 3.15 | 1.04 | 3.39 | 0.76 | 3.06 | 0.002 |
| Control | 3.29 | 1.09 | 3.19 | 1.10 | 3.26 | 0.94 | 3.12 | 1.03 | 3.12 | 1.02 | 3.27 | 0.98 | 3.29 | 1.14 | 3.24 | 1.09 | 3.04 | 0.95 | 1.58 | 0.126 |
| Remote work disadvantages | 2.66 | 0.94 | 2.44 | 0.89 | 2.71 | 0.89 | 2.59 | 0.89 | 2.74 | 0.91 | 2.72 | 0.82 | 2.57 | 0.98 | 2.55 | 0.94 | 2.86 | 0.95 | 4.70 | 0.000 |
| Remote work advantages | 3.64 | 1.00 | 3.69 | 0.94 | 3.66 | 0.87 | 3.62 | 0.92 | 3.56 | 0.95 | 3.70 | 0.87 | 3.72 | 0.92 | 3.69 | 0.97 | 3.49 | 1.08 | 1.38 | 0.202 |
| Emotional exhaustion | 2.70 | 0.78 | 2.64 | 0.82 | 2.83 | 0.63 | 2.81 | 0.76 | 2.83 | 0.74 | 2.80 | 0.70 | 2.79 | 0.84 | 2.75 | 0.78 | 2.80 | 0.65 | 2.26 | 0.021 |

**Table 3.** ANOVA post hoc tests for the TAS sample by work areas.

|  |  |  | 95% CI |
|---|---|---|---|
| Cognitive demands | Logistics and maintenance | Finance | −0.70, −0.09 |
|  |  | Legal affairs | −1.00, −0.02 |
|  | Educational services | Research | −0.28, −0.01 |
|  |  | Finance | −0.55, −0.12 |
|  |  | Legal affairs | −0.88, −0.00 |
|  | Finance | Administration | 0.08, 0.53 |
|  |  | Human resources | 0.01, 0.51 |
| Off-TAJD | Logistics and maintenance | Human Resources | 0.30, 1.11 |
|  |  | ICT | 0.04, 0.84 |
|  |  | Educational services | 0.15, 0.86 |
|  |  | Internationalization | 0.34, 1.26 |
|  |  | Finance | 0.07, 0.96 |
|  |  | Administration | 0.21, 0.95 |
|  | Research | Human Resources | 0.23, 0.78 |
|  |  | Educational services | 0.11, 0.50 |
|  |  | Internationalization | 0.25, 0.95 |
|  |  | Administration | 0.16, 0.60 |
| Workaholism | ICT | Logistics and maintenance | −0.55, −0.02 |
|  |  | Research | −0.41, −0.06 |
|  |  | Educational services | −0.39, −0.06 |
|  |  | Finance | −0.56, −0.04 |
|  |  | Administration | −0.39, −0.03 |
| Relaxation | ICT | Research | 0.08, 0.57 |
|  |  | Educational services | −0.22, 0.49 |
| Mastery | Educational services | ICT | −0.46, −0.04 |
|  |  | Internationalization | −0.56, −0.00 |
| Remote work disadvantages | Educational services | ICT | 0.11, 0.50 |
|  |  | Administration | 0.05, 0.35 |
| Emotional exhaustion | Educational services | ICT | 0.01, 0.36 |

Regarding cognitive demands, logistics and maintenance had the lowest scores, with significant differences from finance and legal affairs, the latter having the highest scores. Finance, legal affairs, and research also showed significantly higher scores than educational services; in addition, finance showed significant differences compared to administration and human resources.

ICT reported lower scores of workaholism compared to almost all other areas, with the exception of legal affairs and internationalization, for which there were no significant differences, likely due to the smaller size of the group.

For off-TAJD, logistics and maintenance reported higher levels than all other areas, with the exception of legal affairs and research, for which there were no significant differences; the latter reported an even higher level, significantly different from human resources, educational services, internationalization, and administration.

Looking at the two recovery experiences that showed significant differences, for relaxation, ICT had higher scores than any other group, with the only significant difference being with research and educational services, reporting the lowest levels; a similar scenario concerned mastery, where educational services reported the lowest scores, with significant differences compared to ICT and internationalization.

Finally, educational services showed higher scores of remote work advantages compared to ICT and administration, and higher scores of emotional exhaustion compared to ICT.

### 3.2. Academics Subsample

Table 4 shows the means and standard deviations of the variables for the male and female academic subsamples, as well as the results from ANOVA. Given the low scale reliability of cognitive demands, the results for this variable are not discussed. With the exception of the recovery experience of control, all variables showed statistically significant differences between women and men. Women reported lower scores for all recovery experiences and higher workload, while at the same time reporting higher scores for both the advantages and disadvantages of remote work. Regarding negative outcomes, women reported higher scores on workaholism, emotional exhaustion, and WFC.

**Table 4.** Means, standard deviations, and ANOVA for the academics sample by gender.

| | Women | | Men | | | |
|---|---|---|---|---|---|---|
| | **M** | **SD** | **M** | **SD** | ***F*** | ***p*** |
| Workload | 3.79 | 0.95 | 3.62 | 0.98 | 15.85 | 0.000 |
| Cognitive demands | 4.32 | 0.68 | 4.18 | 0.69 | 24.07 | 0.000 |
| Off-TAJD | 3.92 | 0.88 | 3.68 | 0.93 | 39.01 | 0.000 |
| Workaholism | 2.77 | 0.79 | 2.56 | 0.79 | 38.82 | 0.000 |
| WFC | 2.89 | 0.85 | 2.68 | 0.84 | 31.11 | 0.000 |
| Detachment | 2.02 | 0.98 | 2.11 | 0.99 | 4.41 | 0.036 |
| Relaxation | 2.87 | 1.04 | 3.16 | 0.97 | 45.84 | 0.000 |
| Mastery | 2.86 | 0.94 | 3.05 | 0.86 | 24.27 | 0.000 |
| Control | 2.81 | 1.03 | 2.75 | 0.99 | 2.06 | 0.151 |
| Remote work disadvantages | 2.81 | 0.86 | 2.70 | 0.83 | 9.26 | 0.002 |
| Remote work advantages | 3.08 | 0.99 | 2.97 | 0.96 | 6.88 | 0.009 |
| Emotional exhaustion | 2.92 | 0.74 | 2.72 | 0.72 | 42.56 | 0.000 |

Means, standard deviations, and ANOVA results for academic levels are presented in Table 5, while post hoc tests are shown in Table 6. All differences between groups are statistically significant. RU showed lower scores of workload, but also higher scores of detachment.

**Table 5.** Means, standard deviations, and ANOVA for the academics sample by role.

| | FP | | AP | | RU | | RTD-B | | RTD-A | | | |
|---|---|---|---|---|---|---|---|---|---|---|---|---|
| | **M** | **SD** | **M** | **SD** | **M** | **SD** | **M** | **SD** | **M** | **SD** | ***F*** | ***p*** |
| Workload | 3.73 | 0.99 | 3.76 | 0.94 | 3.31 | 1.01 | 3.89 | 0.85 | 3.74 | 0.94 | 16.71 | 0.000 |
| Cognitive demands | 4.18 | 0.71 | 4.29 | 0.67 | 4.16 | 0.75 | 4.31 | 0.59 | 4.30 | 0.65 | 4.46 | 0.001 |
| Off-TAJD | 3.83 | 0.86 | 3.85 | 0.92 | 3.59 | 1.00 | 3.87 | 0.85 | 3.73 | 0.92 | 5.58 | 0.000 |
| Workaholism | 2.55 | 0.81 | 2.73 | 0.77 | 2.44 | 0.80 | 2.81 | 0.80 | 2.88 | 0.78 | 15.90 | 0.000 |
| WFC | 2.71 | 0.85 | 2.85 | 0.84 | 2.63 | 0.85 | 2.83 | 0.80 | 2.87 | 0.84 | 5.64 | 0.000 |
| Detachment | 2.05 | 0.97 | 2.05 | 1.02 | 2.30 | 1.04 | 1.90 | 0.82 | 2.03 | 0.92 | 6.46 | 0.000 |
| Relaxation | 3.14 | 1.01 | 2.95 | 1.01 | 3.16 | 1.05 | 2.86 | 1.01 | 2.98 | 0.97 | 5.72 | 0.000 |
| Mastery | 3.01 | 0.87 | 2.97 | 0.92 | 3.10 | 0.93 | 2.77 | 0.89 | 2.77 | 0.86 | 7.36 | 0.000 |
| Control | 2.90 | 1.06 | 2.73 | 1.02 | 2.90 | 1.01 | 2.62 | 0.86 | 2.68 | 0.90 | 5.39 | 0.000 |
| Remote work disadvantages | 2.64 | 0.85 | 2.83 | 0.84 | 2.66 | 0.89 | 2.81 | 0.80 | 2.83 | 0.80 | 5.87 | 0.000 |
| Remote work advantages | 2.89 | 0.99 | 3.01 | 0.98 | 3.09 | 0.99 | 3.16 | 0.96 | 3.19 | 0.92 | 5.34 | 0.000 |
| Emotional exhaustion | 2.68 | 0.76 | 2.86 | 0.73 | 2.82 | 0.73 | 2.81 | 0.73 | 2.81 | 0.73 | 7.21 | 0.000 |

FP and RU both reported significantly lower scores of workaholism only compared to other groups, but not between them; RU also showed the lowest scores of off-TAJD compared to all other groups, though the difference with RTD-A was not significant.

RTD-A reported the highest scores of WFC, significantly different from RU, which reported the lowest value; the other significant differences were found between both FP and RU and AP, which reported the second highest level of WFC

**Table 6.** ANOVA post hoc tests for the academics sample by role.

| | | | 95% CI |
|---|---|---|---|
| Workload | RU | FP | −0.62, −0.23 |
| | | AP | −0.63, −0.28 |
| | | RTD-B | −0.80, −0.37 |
| | | RTD-A | −0.68, −0.18 |
| Cognitive demands | AP | FP | 0.01, 0.22 |
| | | RU | 0.00, 0.27 |
| Off-TAJD | RU | FP | −0.43, −0.05 |
| | | AP | −0.44, −0.09 |
| | | RTD-B | −0.49, −0.06 |
| Workaholism | FP | AP | −0.30, −0.06 |
| | | RTD-B | −0.43, −0.09 |
| | | RTD-A | −0.51, −0.15 |
| | RU | AP | −0.42, −0.14 |
| | | RTD-B | −0.55, −0.18 |
| | | RTD-A | −0.63, −0.23 |
| WFC | AP | FP | 0.01, 0.27 |
| | | RU | 0.07, 0.37 |
| | RU | RTD-A | −0.45, −0.03 |
| Detachment | RU | FP | 0.06, 0.45 |
| | | AP | 0.07, 0.44 |
| | | RTD-B | 0.19, 0.62 |
| | | RTD-A | 0.04, 0.52 |
| Relaxation | FP | AP | 0.03, 0.34 |
| | | RTD-B | 0.06, 0.50 |
| | RU | AP | 0.021, 0.38 |
| | | RTD-B | 0.06, 0.54 |
| Mastery | RTD-B | FP | −0.44, −0.05 |
| | | AP | −0.38, −0.02 |
| | | RU | −0.55, −0.13 |
| | RTD-A | FP | −0.45, −0.03 |
| | | RU | −0.56, −0.11 |
| Control | FP | AP | 0.01, 0.32 |
| | | RTD-B | 0.08, 0.48 |
| | | RTD-A | 0.00, 0.44 |
| | RTD-B | RU | −0.50, −0.06 |
| Remote work disadvantages | AP | FP | 0.06, 0.32 |
| | | RU | 0.02, 0.32 |
| Remote work advantages | FP | RU | −0.39, −0.011 |
| | | RTD-B | −0.48, −0.06 |
| | | RTD-A | −0.52, −0.07 |
| Emotional exhaustion | FP | AP | −0.29, −0.07 |
| | | RTD-A | −0.45, −0.11 |

Regarding relaxation, RU still showed the highest score, but with significant differences only for AP and TRB, which again both differed significantly from FP, which showed comparable scores of relaxation to RU. As for mastery, RTD-A and RTD-B showed the lowest scores, with significant differences from RU and FP, although only RTD-B showed significant differences from AP. Predictably, FP had a high score in control, which was significantly different from the scores of all other roles with the exception of RU, which reported an even higher score, but only significantly different from RTD-B.

FP reported a lower impact of remote work advantages compared to RU, RTD-A, and RTD-B, with no significant differences from AP, which also reported the second lowest impact, though we did not find significant differences from other groups. Conversely, AP reported a high impact of remote work disadvantages, which was significantly different from FP and RU, which reported the lowest scores.

Finally, regarding emotional exhaustion, FP reported the lowest values, significantly different only from AP and RTD-A; while RU reported the highest scores, no significant difference was found.

## 4. Discussion

The aim of this study was to describe the work in Italian universities for academic and technical and administrative staff during the second COVID-19 lockdown, using the framework of the Job Demands-Resources-Recovery model (Kinnunen et al. 2011) with reference to the process of health degradation and, in particular, emotional exhaustion. The study also addressed the advantages and disadvantages of emergency remote working. Starting from an exploratory perspective aimed at providing a general framework for further specific research, the study examined differences in gender and work area (in the TAS sample) and gender and role (in the academics sample).

As for TAS, the data showed a high level of work demands, but accompanied by recovery scores that were almost always above the middle point of the response scale: some difficulties emerged in terms of psychological detachment. The possibility of recovery should have allowed the restoration of resources spent at work (Sonnentag and Fritz 2007). Emotional exhaustion presented lower scores than the middle point of the scale and even lower scores for work–family conflict, although in both cases the scores are higher in the female subsample. The levels of workaholism and off-TAJD do not seem to be particularly critical, while in the general perception of remote work, the advantages outweigh the disadvantages, despite the overall critical situation.

The data on differences in relation to the TAS' work areas indicated that the measures used were able to identify the specificity of different work areas, which is an important element in studies of this type, while also providing useful insight into which areas seem to share similar characteristics in terms of job demands; in addition, it could prove useful in developing targeted measures for specific areas.

Compared to TAS, the mean scores of the variables in the academics subsample are more critical overall, with higher perceived levels of demands and especially more "additional" work demand through new technologies (off-TAJD); this high-demand work environment is associated with greater difficulty in recovery.

Thus, how can we explain the more critical scores in the female subsamples, in both TAS and academics samples? We can assume that women were simultaneously more engaged on the family front, considering the persistence of the traditional family model in Italy, with an unbalanced distribution of the care burden between men and women (Dolce et al. 2020; Saraceno 2013; Addabbo et al. 2012), especially in relation to childcare, which, globally, was particularly demanding during lockdowns (Fisher et al. 2020; Moreira da Silva 2019; Power 2020; Pozzan and Cattaneo 2020). More generally, other studies have documented more problematic outcomes for women during the pandemic (Gualano et al. 2020; Liu et al. 2020), but scholars had already shown gender differences in these indicators prior to the pandemic, which were associated with greater commitment to work–family balance and less chance of recovery (Purvanova and Muros 2010; Nolen-Hoeksema and Harrell 2002).

Our research confirms that these differences were strongly evident in Italian academia during the second lockdown. It is one of the first systematic studies to provide comprehensive evidence of these phenomena within universities, albeit in a particular situation such as remote work in an emergency situation. It also supports some of the assumptions about women's greater difficulties in their academic careers due to higher levels of family

and academic domestic work (Heijstra et al. 2016, 2017). Mixed-method studies should investigate the nature and dynamics of this overload.

The differences we observe in terms of academic roles reveal more problematic experiences for the first stable academic career roles, which are still precarious positions, and for Associate Professors, i.e., roles that are still fully engaged in the dynamics of career confirmation, in a highly competitive context that more or less implicitly demands high levels of time and energy commitment even outside "typical" working hours, as evidenced by the reported high levels of off-TAJD. Indeed, the opportunity for professional affirmation is associated with a complexity of tasks and positive results in terms of research, teaching, public engagement, and participation in the functioning of the institution (Agasisti and Soncin 2021). These are also the academic positions in which we find women and men taking on childcare duties, when they are not simultaneously caring for their own children and their aging parents (the well-known phenomenon of the sandwich generation; Brenna 2021).

To understand the less problematic outcomes of Full Professors and RU, we can refer to both the academic career process and the characteristics of these roles. As is well known, the position of Full Professor in Italy is predominantly "male", the average age is high, the domestic care duties are less, and the salary high; in most cases, there is a working group to support the various research and teaching activities. As mentioned in the description of academic positions in Italy, there have been no new RU hires since the reform. In recent years, a significant part of this population has gained access to the role of Associate Professor through dedicated selection procedures. Those who still remain in this role might experience academic competition with more detachment or resignation (Guidetti et al. 2020), with less activation and workload, which is also allowed by the reference legislation for this role (low commitment to teaching, less involvement in institutional roles). One piece of information we do not have concerns the possible choice of part-time employment, a condition that allows to perform other extra-university activities, which, combined with a lower workload, can have positive effects as an outcome of multi-tenure in a process of general enrichment of resources.

*Limitations*

Although this study includes a large sample and covers the entire national territory, it is not without limitations. First, it is a cross sectional and descriptive study, which limits the scope of our conclusions. The second limitation is the low response rate, which is unfortunately quite common in surveys carried out in universities, especially concerning academics. Third, as mentioned above, the indicators cannot be compared with previous descriptions, due to the lack of similar studies in Italian universities because of the exceptional and unprecedented nature of MWFH. Finally, we must consider the low reliability of the scale measuring cognitive demands, though only for the academic subsample. This could be due to the small numbers of items, a decision taken to ensure that the questionnaire was reasonably lean in order to maximize the expected low response rate.

**5. Conclusions**

This study starts from the observation that it is important to have systematic results on the variables considered in the present study, as a reference for the development of policies and interventions. The systematic conduct of such studies and their visibility is fundamental to build a system of interventions based on empirical evidence and to set in motion good practices of virtuous cycles between research and intervention.

Italian law requires companies (including universities) to assess stress but allows for different methods that leave much room for improvisation and do not allow for consideration at a general, national level. Even if the possibility to customize surveys on these topics is fundamental from the perspective of situated organizational analysis (Gorli et al. 2009), it is also important to have some general observatories that allow placing specific data in a broader frame of reference. Thus, it is a matter of promoting evaluation processes

with mixed methods that use general, reliable, and valid measures, specific aspects, and qualitative findings.

Analyzing work dynamics in academia, taking into account gender differences, is a necessary prerequisite for defining action plans to promote well-being and gender equality. This analysis requires a situated approach that is able to take into account the specific field of study, i.e., science as a social institution (Picardi 2020) and, even more specifically, a particular university as a place with its own uniqueness, characterized by its own history, elements of the present, and perspectives that must be considered when designing interventions. While a particular university may share structural, procedural, regulatory, and cultural elements with other research institutions in the same country, it may also have its own unique characteristics that result from its history, successive governance structures, and perspectives that partly determine future trends.

In this sense, "research for action" approaches, such as the present, are fundamental. Starting from a general approach, they can later allow specific universities to carry out ad hoc studies capable of analyzing the phenomena in depth and thus proceed to the definition of actions. This approach can promote sustainable working conditions that can promote greater well-being and improved quality in higher education, consistent with Goal 3 and 4 of the 2030 Agenda (Gamage et al. 2022).

Even if they remain at a general level of analysis of the national situation, further studies are certainly necessary to understand the relationships between the variables studied, in all their complexity, and to integrate a qualitative evaluation of the results, possibly with the help of focus groups to discuss the main findings. It is not only important to understand the current dynamics, but also to evaluate the impact of the measures introduced and, even before, their actual use.

Finally, to be truly consistent with Sustainable Development Goal 5, it is essential to reiterate that there is still a large gap between theoretical considerations, empirical research, and the actual implementation of gender equality measures (Verloo 2013; De Vries and Brink 2016), in the illusion, unsupported by evidence, that the meritocracy constructed by reform processes is in and of itself gender neutral. Building adequate psychosocial research systems for the academy provides essential contextual data for developing gender equality plans, which are now mandatory to participate in Horizon calls and, along with gender budgeting, to access recovery and resilience funds for research. Moreover, they could serve as the basis for building and reporting strategic planning.

**Author Contributions:** Conceptualization, C.G., D.S., T.A., V.B., R.M., I.P., P.T., G.G. and D.C.; Methodology, C.G. and D.S.; Formal analysis, D.S.; Investigation, C.G., T.A., V.B., R.M., I.P., P.T., G.G. and D.C.; Data curation C.G. and D.S.; Writing—original draft preparation, C.G. and D.S.; Writing—review & editing, C.G., D.S., T.A., R.M., I.P., P.T., G.G. and D.C.; Supervision, C.G.; Project administration, C.G., D.S. and T.A. All authors have read and agreed to the published version of the manuscript.

**Funding:** This research received no external funding.

**Institutional Review Board Statement:** The study was conducted according to the guidelines of the Declaration of Helsinki and approved by the bioethics committee of the University of Turin on 22/10/2020 (Prot. No. 458997).

**Informed Consent Statement:** Informed consent was obtained from all subjects involved in the study.

**Data Availability Statement:** The data presented in this study are not publicly available due to the Italian privacy law. The data are available on request from the corresponding author.

**Acknowledgments:** The authors would like to thank Sara Colombini, Eleonora Costantini and Federica Pillo for their organizational support and the Guarantee Committees—participating in the National Conference—for their contribution in disseminating the research in their respective universities.

**Conflicts of Interest:** The authors declare no conflict of interest.

## Note

[1] Although terms may differ between countries, for the sake of simplicity and convenience, in this paper we used "academics" for teaching and research staff and "TAS" for technical and administrative staff, i.e., non-academics.

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
