# Peer review of "The Show Must Go On: A Snapshot of Italian Academic Working Life during Mandatory Work from Home through the Results of a National Survey"

_socsci, doi:10.3390/socsci11030111_

Round 1

Reviewer 1 Report

The work deals with a very important issue, which is academic working life during mandatory work from home. The title of the article is great; it attracts attention, which encouraged me to apply for a review of the work. The authors analyzed the literature very well, which is undoubtedly the strong point of the article. In the introduction, the authors do not specify the purpose of the work. They immediately move on to the next point, where they analyze the literature. The introduction should be refined. It is a very important point that allows readers to understand further research.

Moreover than authors write about the aim in coclusion: "The aim of this study was to describe the work in Italian universities for academic
and technical and administrative staff during the second COVID-19 lockdown., using the framework of the Job Demands-Resources-Recovery model  with reference to the process of health degradation and, in particular, emotional exhaustion. " Lack of figure 1 in text. The authors did not create hypotheses, which significantly reduces the value of the work, which is a pity because the material is very interesting. Very well described respondents. Methodology well presented and described.

Author Response

Thank you very much for your comments. We proceeded to modify the paper as per your suggestions.

Introduction and aim of the study: We added a passage in the introduction which describes the aim of the study in more detail. We also left the aim in the conclusion in order to summarize the paper.

Lack of figure 1 in text: We added the figure back.

Reviewer 2 Report

Congratulation. Very interesting topic at the moment and good work. The Job Demand-Resource-Recovery Model proved to be useful.

Correct minor typos mistakes: first word in page 4 "neithe". There are a few more errors.

Author Response

Thank you very much for your comments. We proceeded to correct the typos.

Reviewer 3 Report

This paper pertains to the psychological state of Italian academic organization employees as it was reported during the current pandemic. The paper offers descriptive statistics for various subgroups, and mostly offers comparisons about relative magnitudes of stress. I will leave the issue of whether a paper on the topic is suitable for this journal to the editors. I suspect they will believe that it is.

My first note pertains to the title, which I believe a poor choice for the paper. I was expecting something qualitative, which the paper is definitely not. Something more descriptive about the survey or about the model would be better.

The paper is quite limited by the lack of a pre-pandemic benchmark. I am unsure what to make of the levels that are being reported without a “before.” Whereas it is easy to blame everything on the forced changes due to COVID-19, that would be unjust. Most studies suggest that stress is always high and everywhere.

A main thrust of the paper pertains to gender equality. What the paper purports to offer would be a surprise to few. However, I would credit the authors with its strong documentation with a large sample size.

I wish that the paper had done less. In addition to gender, we have faculty vs. staff comparisons across academic areas and for different academic ranks, all for a variety of psychological overload dimensions. It gets to be too much at times.

I conclude that there might be value in a primarily descriptive analysis of data collected by another group. The paper is slow to develop and could have been delivered much more parsimoniously.

Author Response

Thank you very much for your comments, we have done some modifications following your notes.

Title: two out of three reviewers rated the title positively, which we intend to leave as it is with the addition of a small change ("though the results of a national survey") to better convey the quantitative nature of the study.

Pre-pandemic benchmark: The paper highlights the relevant, pre-pandemic literature, (which is why the literature review section is so long and, as you noted, slows the development of the paper), pointing out that stress levels were already problematic. This issue is further highlighted in this passage.

Gender equality: We agree with your comment, findings are not surprising, but there is little systematic empirical evidence on the matter concerning the Italian academic context, compared to what personal experiences or local reports tell us. Having systematic empirical data also supports actions for change. 

Too many variables: we understand that the paper can be too broad at times, however its aim it's indeed to offer a first systematic outlook of the current situation, which is lacking in existing literature.